# The Organic Ammonium Counterion Effect on Slow Magnetic Relaxation of the [Er(hfac)$_4$]$^−$ Complexes

Tatiana G. Prokhorova, Denis V. Korchagin *, Gennady V. Shilov, Alexei I. Dmitriev, Mikhail V. Zhidkov and Eduard B. Yagubskii *

Federal Research Center of Problems of Chemical Physics and Medicinal Chemistry, Russian Academy of Sciences, 142432 Chernogolovka, Russia; prokh@icp.ac.ru (T.G.P.); genshil@icp.ac.ru (G.V.S.); alex-dmitriev2005@yandex.ru (A.I.D.); zhidkov@icp.ac.ru (M.V.Z.)
* Correspondence: korden@icp.ac.ru (D.V.K.); yagubski@icp.ac.ru (E.B.Y.)

**Abstract:** The first mononuclear anionic erbium complex [Er(hfac)$_4$]$^−$ (hfac = hexafluoroacetylacetone) with an organic ammonium cation [(CH$_3$)$_4$N$^+$] as the counterion was synthesized and structurally and magnetically characterized. The coordination geometries around the Er ions are square antiprisms with pseudo-$D_{4d}$ symmetry. The complex shows distinct field-induced slow magnetization relaxation, which is described by a combination of Orbach (U$_{eff}$/k$_B$~28.54(8) K) and direct mechanisms. Quantum chemical calculations were performed to analyze the magnetic properties of the complex under consideration.

**Keywords:** mononuclear complexes of rare earth elements with β-diketone ligands; ligand environment geometry; single-molecule magnets (SMMs); slow magnetic relaxation; anisotropy barrier

## 1. Introduction

Currently, the most actively developing direction in the field of single-molecule magnets (SMMs) is associated with the synthesis and study of mononuclear complexes of rare earth elements, the so-called single-ion magnets (SIMs) [1,2]. Unlike 3D metals, lanthanide ions have a strong spin–orbit coupling, which determines their significant uniaxial magnetic anisotropy, the enhancement of which is an effective strategy for increasing the magnetization reversal barrier (U$_{eff}$) and the blocking temperature (T$_B$) [3–6]. Exceptionally high SMM characteristics, such as record values of U$_{eff}$ = 1541 cm$^{−1}$ and T$_B$ = 80 K, are recorded for the sandwich metallocene complex of dysprosium [(Cp$^{iPr5}$)Dy(Cp*)]$^+$ at present [7]. For SIMs, magnetic anisotropy is determined by two factors: the internal anisotropy of the paramagnetic metal center and the ligand field. Control of coordination geometry and symmetry around lanthanide ions is important for achieving high SMM characteristics [8]. The geometry of the ligand environment can enhance the local magnetic anisotropy [9]. Strengthening the coordination environments of Ln ions, such as $D_{nd}$ and $D_{nh}$ (where $n$ = 4–6), was found to be effective in suppressing the rapid magnetization relaxation associated with quantum tunneling [10–14].

The β-diketone ligands (acetylacetone, dibenzoylmethane (dbm) and their derivatives) have been actively used to create the appropriate local symmetry of $D_{4d}$ in mononuclear Ln(III) complexes. Most of these complexes are neutral and contain mixed ligands: three monoanionic β-diketone ligands and two other neutral ligands or one bidentate neutral ligand per Ln(III) ion [15–24], whereas anionic SMMs containing the [Ln(β-diketone)$_4$]$^−$ structural motif are rare [25–29]. Using hexafluoroacetylacetonate (hfac) as a ligand, four anionic [Ln(hfac)$_4$]$^−$ complexes with Cs and K counterions were synthesized: [Cs{Dy(hfac)$_4$}] (**1**), [Cs{Er(hfac)$_4$}] (**2**), [K{Dy(hfac)$_4$}] (**3**), and [K{Er(hfac)$_4$}] (**4**) [25]. The compounds have a chain structure in which the atoms of lanthanides and alkali metals are alternately linked by the hfac ligands. Each lanthanide atom in the chain is coordinated by eight oxygen atoms from four hfac ligands.

The most significant difference between the cesium (**1**, **2**) and potassium (**3**, **4**) salts lies in the coordination geometry of the O8 environment around the lanthanide ion. For compounds with cesium, which has a large ionic radius, the dodecahedral geometry with pseudo-$D_{2d}$ symmetry of the $[Ln(hfac)_4]^-$ anion was detected, whereas a distorted square antiprism with pseudo-$D_{4d}$ symmetry is observed in **3** and **4** when the cation is $K^+$. A study of magnetic properties showed that complexes **3** and **4** are field-induced single-ion magnets in contrast to **1** and **2**. This means that the coordination geometry of the lanthanide ions and, hence, the magnetic relaxation of the anionic $[Ln(\beta\text{-diketone})_4]^-$ complexes can be controlled by the counterion. Note that the magnetization barriers for Dy and Er complexes with the coordinate geometry of the distorted square antiprism are close (23.95 and 20.21 K, respectively), although dysprosium and erbium have fundamentally different shapes of 4f-electron clouds (oblate and prolate, respectively) [30]. This is observed when both the local symmetry and the electron density crystal field distribution satisfy the Dy and Er SIMs simultaneously [31]. Recently, the effect of counterion on the structure and magnetic relaxation of $[Dy(dbm)_4]^-$ complexes with various tetra-alkyl ammonium cations remote from the Dy(III) spin center was studied [28,29]. Magnetic measurements showed that the alkyl chain length of ammonium ions is important in tuning the SMMs' performance.

In contrast to Dy, anionic complexes of Er with β-diketones, whose magnetic behavior has been explored, are known in the literature in solitary examples [25]. Herein we report the synthesis of the first Er anion complex with a hfac ligand containing an organic quaternary ammonium cation as the counterion, $(CH_3)_4N[Er(hfac)_4]$ (**5**). The crystal structure and magnetic properties of the complex were investigated.

## 2. Materials and Methods

### 2.1. Synthesis

All starting reagents were used as received: $Er(CF_3SO_3)_3$ (Aldrich); Hhfac (Acros organics); and $N(CH_3)_4OH$ solution (Sigma-Aldrich, St. Louis, MO, USA). The compound $(CH_3)_4N[Er(hfac)_4]$ (**5**) was obtained by a procedure similar to that described in [25]. The exact conditions for the synthesis are the following:

The mixture of 0.57 mL of Hhfac (0.832 g, 4 mmol) and 1.68 mL of a 25% methanol solution of $N(CH_3)_4OH$ (0.364 g, 4 mmol) in 7.5 mL of methanol was added to a hot solution of $Er(CF_3SO_3)_3$ (0.614 g, 1 mmol) in 10 mL of methanol. The resulting solution was refluxed for 3 h with stirring. After that, the resulting pink solution was cooled to room temperature, filtered and left to stand for slow evaporation. Light pink plate-like crystals suitable for single-crystal X-ray analysis were isolated after 3 weeks. Yield: 35%. According to X-ray electron probe microanalysis (EPMA) data, the ratio of F:Er atoms for $C_{24}H_{16}F_{24}O_8NEr$ (compound **5**) is approximately 23.9:1; IR (cm$^{-1}$): 1649, 1602, 1559, 1532, 1510, 1485, 1350, 1251, 1202, 1131, 1097, 950, 798, 767, 741, 660, 585, 528 and 470.

### 2.2. Electron-Probe X-ray Microanalysis

The electron-probe X-ray microanalysis (EPMA) of obtaining crystals was performed on a JEOL JSM-5800L scanning electron microscope (SEM) at a 100-fold magnification and an electron beam density of 20 keV. The depth of beam penetration into the sample was 1–3 μm.

### 2.3. The IR Spectra

The IR spectra were recorded on a Bruker ALPHA Fourier spectrometer (Bruker Optik GmbH, Ettlingen, Germany) in the frequency range 400–4000 cm$^{-1}$ in the attenuated total reflectance mode (FTIR–ATR).

### 2.4. X-ray Diffraction

X-ray diffraction analysis of compound **5** was carried out on a CCD Agilent XCalibur diffractometer with an EOS detector (Agilent Technologies UK Ltd., Yarnton, Oxfordshire, UK). Data collection, determination and refinement of unit cell parameters were carried out using the CrysAlis program [32]. X-ray diffraction data at 100 K for compound **5** were col-

lected using $MoK_{\alpha}$ ($\lambda = 0.71073$ Å) radiation. The structure was solved by the direct method. The positions and thermal parameters of non-hydrogen atoms were refined anisotropically by the full-matrix least-squares method. The positions of the hydrogen atoms were refined with riding model constraints. The X-ray crystal structure data were deposited with the Cambridge Crystallographic Data Center with reference code CCDC 2263014. All calculations were performed with the SHELXTL program package [33]. The crystallographic parameters and the data collection and refinement statistics are summarized in Table 1. Selected bond lengths and angles are given in Table S1.

**Table 1.** Crystal data and structure refinement for $(CH_3)_4N[Er(hfac)_4]$.

| Parameter | Value |
|---|---|
| Empirical formula | $C_{24}H_{16}Er_1F_{24}N_1O_8$ |
| Formula weight | 1069.64 |
| Temperature, K | 100.0(1) |
| Wave length | 0.71073 Å |
| Crystal system, space group | Monoclinic, $P2_1/c$ |
| $a$, Å | 28.1245(3) |
| $b$, Å | 18.1069(2) |
| $c$, Å | 20.7861(2) Å |
| $\beta$, deg | 91.146(1) |
| Volume, Å$^3$ | 10,583.2(2) |
| Z, Calculated density, Mg/m$^3$ | 12, 2.014 |
| Absorption coefficient, mm$^{-1}$ | 2.551 |
| F(000) | 6180 |
| Crystal size, mm | $0.20 \times 0.15 \times 0.05$ |
| Theta range for data, deg | 2.860 to 29.617 |
| Reflections collected/unique | 53,795/24,875 [R(int) = 0.0275] |
| Completeness to $\theta = 25.2$, % | 99.8% |
| Goodness-of-fit on F$^2$ | 1.047 |
| Final R indices [I > 2σ(I)] | R1 = 0.0396, wR2 = 0.0732 |
| R indices | R1 = 0.0671, wR2 = 0.0827 |
| Largest diff. peak and hole, e·A$^{-3}$ | 1.835 and −2.031 |
| CCDC | 2,263,014 |

The powder XRD pattern for **5** was recorded at 295 K on an Aeris diffractometer (Malvern PANalytical B.V., Almelo, The Netherlands). The powder XRD measurements showed that polycrystalline sample **5** is a monophase product corresponding to the single crystal data (Figure S1, ESI).

*2.5. Magnetic Measurements*

Direct current (DC) and alternating current (AC) magnetic properties of complex **5** were studied at a vibrating sample magnetometer of a Cryogen Free Measurement System (CFMS, Cryogenic Ltd., London, UK). The temperature dependence of the magnetic moment M(T) was measured at T = 2–300 K in a DC magnetic field B = 0.5 T. The field dependences of the magnetic moment M(B) were obtained at temperatures 2, 3 and 5 K in the DC magnetic field range of 0–5 T. The AC magnetization of complex **5** was studied at T = 3–10 K in a 4 Oe oscillating field with a frequency range of 10–1000 Hz in the absence and with the application of a DC magnetic field of B = 0.1 T. All experiments were carried

out on a sample mixed with mineral oil (Fomblin YR 1800) to avoid the orientation of individual powder crystallites in a constant magnetic field. Powder sample **5** was sealed in a sample holder (a polyethylene unit). The experimental data were corrected for the sample holder and mineral oil. The diamagnetic contribution from the ligand was calculated using Pascal's constants.

### 2.6. Quantum Chemical Calculations

CASSCF/RASSI + SO/SINGLE_ANISO calculations for the isolated $[Er(hfac)_4]^-$ complex were performed using the OpenMolcas program [34,35]. All calculations were based on the X-ray structure. The [ANO-RCC...8s7p5d3f2g1h] basis set for Er atoms, the [ANO-RCC...3s2p1d] for F and O atoms, the [ANO-RCC...3s2p] for C atoms, and the [ANO-RCC...2s] for H atoms were employed. The ground state f-electron configuration for Er(III) is $4f^{11}$, with $^4I_{15/2}$ multiplet as the ground state. At first, the guess orbitals were generated, from which seven Er-based starting orbitals occupied by eleven electrons were selected to perform the CASSCF calculations. Using the active space involving 35 quartets and 112 doublets, the configuration interaction (CI) procedure was computed. All these 35 quartets and 112 doublets were mixed using the RASSI-SO module to compute the spin–orbit states. The second-order Douglas–Kroll–Hess (DKH) [36–39] Hamiltonian was used to treat the scalar relativistic effects. After computing these spin–orbit states using the SINGLE_ANISO code [40], the corresponding *g*-tensors and the CF parameters for the eight low-lying Kramers doublets (KD) were extracted. The Cholesky decomposition for two-electron integrals is employed throughout the calculations to reduce disk space.

## 3. Results and Discussion

### 3.1. Crystal Structure

Compound $(CH_3)_4N[Er(hfac)_4]$ (**5**) crystallizes in the monoclinic $P2_1/c$ space group. The asymmetric unit contains three $[Er(hfac)_4]^-$ anions and three $[(CH_3)_4N]^+$ cations in the general site. The general view of the asymmetric part of the crystal structure of the $(CH_3)_4N[Er(hfac)_4]$ complex is shown in Figure 1.

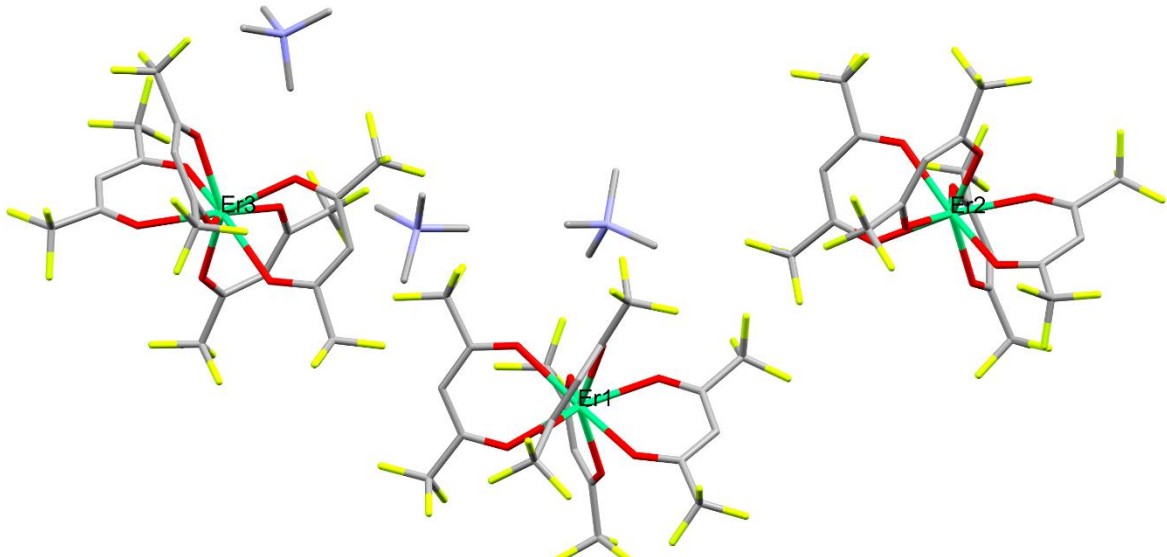

**Figure 1.** The asymmetric part of the crystal structure of $(CH_3)_4N[Er(hfac)_4]$ (**5**). H atoms are omitted for clarity.

The $[Er(hfac)_4]^-$ anion is a mononuclear complex, in the structure of which the Er(III) ion is eight-coordinated with hfac-oxygen's atoms and possesses a distorted square antiprism (pseudo-$D_{4d}$) geometry according to the *SHAPE* analysis [41,42] (Table S2). The nearest coordination environment of Er(III) ions is shown in Figure 2. The coordination

geometry of hfac-oxygen's atoms around the Er(III) ion is somewhat different from each other, but the type of the Er(III) environment is the same.

**Figure 2.** The coordination environment of the Er(III) ions. Dashed lines connect oxygen atoms at the base of square antiprisms. The bond lengths Er–O are indicated.

The average Er–O bond lengths for all symmetry non-equivalent Er complexes are close to each other and equal to 2.320(3), 2.320(3) and 2.322(3) Å, respectively, for Er1, Er2 and Er3. O−Er−O angles lie also in close ranges 71.9(1)−149.2(1), 70.1(1)−147.9(1) and 71.1(1)−148.6(1), respectively (Table S1, ESI). A comparison of other parameters characterizing the square antiprismatic (pseudo $D_{4d}$) coordination environment of crystallographically independent Er ions also indicates their small differences from each other. Despite the strong scatter of values (Figure 2), the average distance between the four neighboring oxygen atoms in the basal planes of SAP $d_{in}$ is 2.760, 2.767 and 2.772 Å (for Er1, Er2 and Er3, respectively), and the interplanar distances ($d_{pp}$) are 2.517, 2.499 and 2.488 Å between the upper and lower planes. The basal planes of SAP are nearly parallel, with a slight tilt angle in the range of 1.3–2.4°. The ratio $d_{in}/d_{pp}$ indicates an axial compression of the SAP around the Er(III) ions. The upper and lower bases of the SAP are twisted relative to each other by the smallest angles of 37.2, 39.4 and 39.5° (for Er1, Er2 and Er3, respectively). The found angles are not so close to those expected for an ideal $D_{4d}$ symmetry ($\phi = 45°$).

In contrast to [Er(hfac)$_4$]$^-$ compounds with inorganic counterions [25], the shortest intermolecular Er···Er distance is somewhat longer and is 9.67 Å (versus 8.52 for **2** and 7.94 Å for **4**). In the crystal structure of **5**, one can conditionally distinguish layers parallel to the *AC* crystallographic plane (Figure S2), in which anions and cations are bound due to C–H . . . O and C–H . . . F intermolecular interactions. In general, these intermolecular interactions are observed between cations and anions, but one direct C–H . . . F contact between [Er(hfac)$_4$]$^-$ anions was also found. The crystal packing of **5** is stabilized in addition to electrostatic forces and C–H . . . O and C–H . . . F interactions due to van der Waals contacts F . . . F and C . . . F lying in the ranges 2.68–2.93 and 3.14–3.16 Å, respectively (Table S3, ESI).

### 3.2. Magnetic Properties

#### 3.2.1. Direct Current (DC) Magnetic Properties

The DC magnetic susceptibility data for complex **5** were measured in the temperature range of 2.0–300 K under an applied field of 5000 Oe (Figure 3a). The room temperature $\chi_M T$ value of 10.87 cm$^3$ K mol$^{-1}$ is found to be close to the calculated value of 11.48 cm$^3$ K mol$^{-1}$ for non-interacting Er(III) ions ($^4$I$_{15/2}$, S = 3/2, L = 6, J = 15/2, g = 6/5). The experimental data are in good agreement with the dependence obtained on the basis of quantum chemical calculations, with the exception of the low-temperature region (2–10 K). This difference in the experimentally observed value of the $\chi_M T$ can be due to the exchange interaction between Er(III) ions. The non-zero negative value of the Weiss constant of

−10.07 K confirms the presence of antiferromagnetic exchange correlations (Figure S3, ESI). However, the value of the Weiss constant must be treated with caution since it can be caused not only by exchange or dipole interaction but also by the depopulation of the thermally excited levels arising due to the crystal-field splitting. This decrease leads to a negative, non-negligible Weiss constant even if there are no magnetic moment interactions. The field dependences of the magnetic moment *M(H)* almost saturate with the magnetization 3.88 *Nβ* at *T* = 2 K (Figure 3b). This value is noticeably less than the theoretical saturation value of 9 *Nβ* for non-interacting Er(III) ions, which can be explained by the crystal-field splitting effects. The nonsuperimposed M versus H/T curves confirm the presence of magnetic anisotropy (Figure S4, ESI). The saturation magnetization decreases with increasing temperature (Figure S5, ESI). The field dependences of the magnetic moment demonstrate hysteresis loops (Figure S6, ESI) with a coercive field (47 Oe at *T* = 2 K, Figure S5, ESI) depending on the temperature and the magnetic field sweep rate (Figure S5, ESI). The presence of a magnetic hysteresis loop is surprising because, for a zero DC field, there is a fast relaxation of the magnetization, as shown below by the AC measurements. Magnetic hysteresis measurements with different magnetic field sweep rates were performed. It should be noted that the coercive field is almost equal to the scan rate (Figure S6c, ESI). From extrapolation of the dependence of the coercive field on the magnetic field sweep rate to very low sweep rates (0 Oe/s), the coercive field becomes almost zero (Figure S6c, ESI). Thus, the magnetic hysteresis is most likely related to the magnetic field sweep rate.

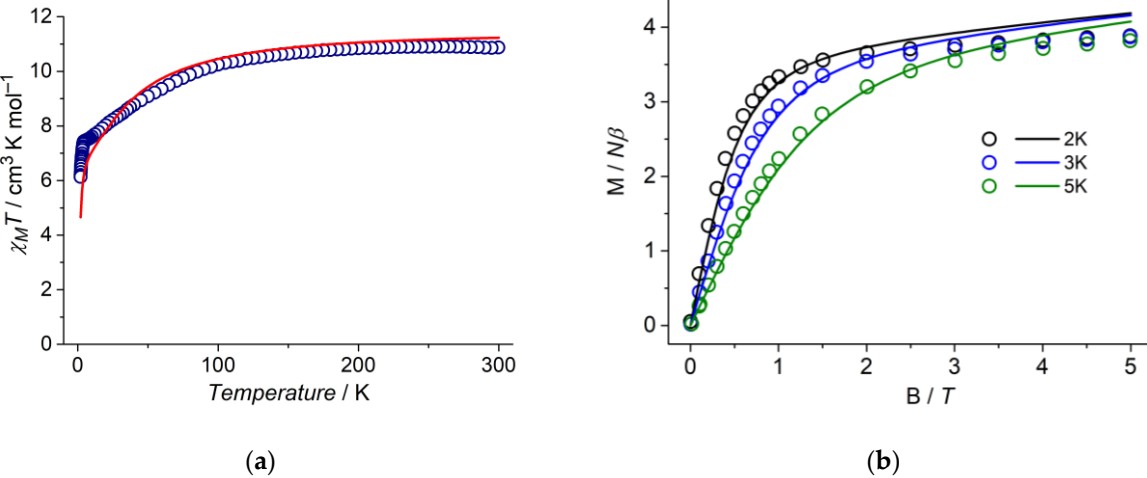

(**a**)                                                          (**b**)

**Figure 3.** (**a**) Temperature dependence of the $\chi_{\mathrm{M}}T$ product; (**b**) Field dependences of the magnetic moment at *T* = 2, 3 and 5 K. (circles—experimental data; solid lines—theoretical estimations based on quantum chemical calculations).

### 3.2.2. Alternating Current (AC) Magnetic Properties

For studying the spin dynamics, frequency-dependent AC susceptibility data were recorded. No peaks are registered for the out-of-phase signals at zero DC field (Figure S7a, ESI). At 4 K in a zero DC field, the out-of-phase signal of magnetic susceptibility $\chi''$ of complex **5** is somewhat enhanced in the frequency interval of 10–10,000 Hz and reaches a maximum at frequencies above 10 kHz, which is beyond the measurement range of our magnetometer (Figure S7a, ESI). The optimal DC magnetic field to suppress partial or complete relaxation associated with quantum tunneling magnetization (QTM) was obtained from the frequency dependencies of AC signals ($\chi'$ and $\chi''$) recorded at constant temperature *T* = 4 K and different DC magnetic fields 0–5000 Oe, as shown in Figure S7a,b, ESI. Maximum values of out-of-phase AC susceptibility $\chi_{\mathrm{M}}''$ and relaxation time $\tau$ were registered in a DC field of 1000 Oe (Figure S7a,d, ESI). Therefore, the frequency dependences of the AC susceptibility at different temperatures of 3–10 K were obtained in this optimal DC field (Figure 4a,b). In a constant field of 1000 Oe, both $\chi'$ and $\chi''$ components of AC susceptibilities show frequency-dependent signals, which indicates that the complex $(CH_3)_4N[Er(hfac)_4]$ demonstrates a slow magnetic

relaxation. With increasing temperature, the $\chi''$ maxima shift to higher frequencies. This behavior of $\chi''$ from frequency is characteristic of SMMs. The Cole–Cole plots ($\chi''$ vs. $\chi'$) presented in Figures 5a and S7c testify to the relaxation processes in **5**.

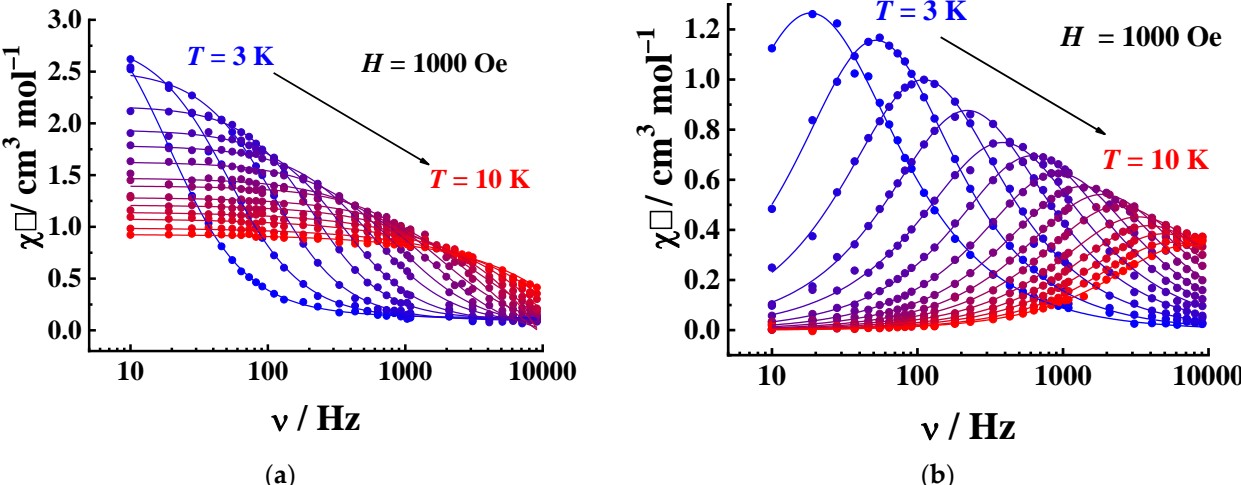

**Figure 4.** Frequency dependences of the in-phase (**a**) and out-of-phase (**b**) AC susceptibility signals at an optimal 1000 Oe DC field and different temperatures.

In contrast to the $[Er(hfac)_4]^-$ complex with the $K^+$ cation, which shows the frequency dependence of $\chi''$ in the DC field of 1000 Oe in a narrow range of temperatures (1.8–3.0 K) and frequencies near 1000 Hz only [25], the complex **5** demonstrates distinct $\chi''$ maxima over a wide range of temperatures (3–10 K) and frequencies (10–10,000 Hz).

The fit of frequency dependences of in-phase $\chi_M'$ and out-of-phase $\chi_M''$ AC susceptibility dependences was performed with Debye functions (Tables S4 and S5, ESI). The dependence of $\ln(\tau)$ vs. inverse temperature for the relaxation process is shown in Figure 5b. The fit of the experimental data was performed with Equation (1), including Orbach and Raman mechanisms.

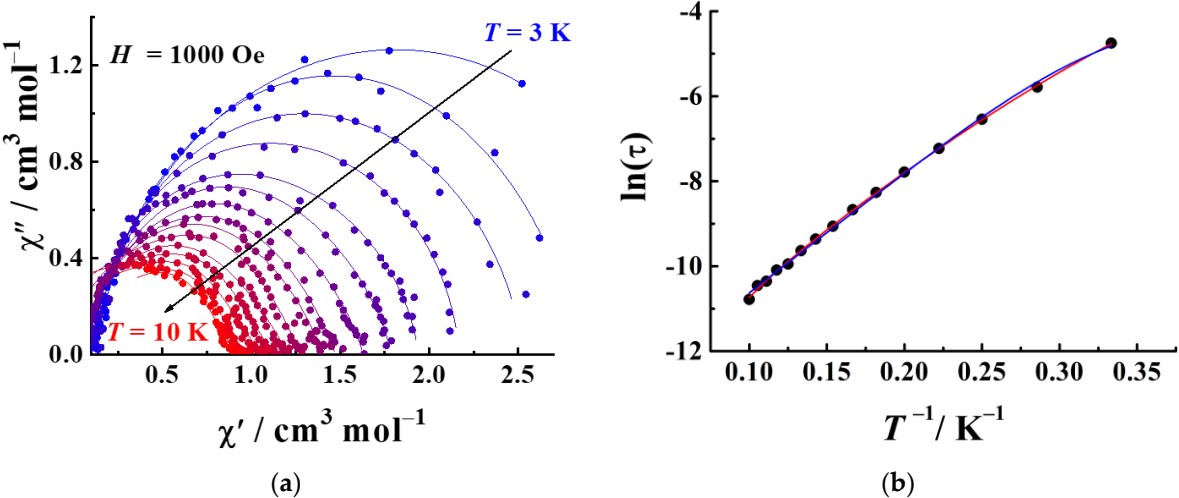

**Figure 5.** (**a**) Cole–Cole plots at 1000 Oe DC fields and indicated temperatures for complex **5**; and (**b**) the $\ln(\tau)$ vs. $1/T$ at the 1000 Oe DC field. Red and blue solid lines are fitted with Equations (1) and (2), respectively.

$$\tau^{-1} = \underbrace{\tau_0^{-1} \exp(-U_{eff}/k_B T)}_{Orbach} + \underbrace{CT^n}_{Raman} \tag{1}$$

Data fitting using Equation (1) gives an energy barrier of 26.86(3) K, with the pre-exponential factor $\tau_0$ of 2.88(9) $\times$ $10^{-6}$ s and $C_{Raman}$ = 0.39(4) $s^{-1} \cdot K^{-n}$, $n_{Raman}$ = 4.75(2). Given the almost linear dependence of the $\ln(\tau)$ vs. inverse temperature fit of the experimental data, it was also performed with Equation (2), including Orbach and direct mechanisms.

$$\tau^{-1} = \tau_0^{-1} \exp(-U_{eff}/k_B T) + AH^2 T$$
$$\qquad\qquad Orbach \qquad Direct$$

(2)

Fit by Equation (2) gives a higher energy barrier of 28.54(8) K with a lower pre-exponential factor $\tau_0$ of 1.40(3) $\times$ $10^{-6}$ s and $A_{Direct}$ = 2.38(8) $\times$ $10^3$ $T^{-2} \cdot s^{-1} \cdot K^{-1}$. As can be seen from Figure 5b, the quality of fits by both Equations (1) and (2) is the same and gives close values of the energy barrier and pre-exponential factor for the Orbach component.

The approximation of the dependence of $\ln(\tau)$ vs. inverse temperature does not allow one to give an unambiguous answer according to the occurrence of the relaxation mechanisms (Raman or Direct). Therefore, a fit of the field dependence of the relaxation time was performed with Equation (3) (Figure S7d, ESI).

$$\tau^{-1} = B_1/(1 + B_2 H^2) + AH^2 T + B_4$$
$$\qquad QTM \qquad\quad Direct \quad Orbach\ and\ Raman$$

(3)

where the first two terms correspond to the field-dependent quantum tunneling of magnetization (QTM) and direct mechanisms of magnetic relaxation. The third term corresponds to the field-independent Orbach and Raman magnetic relaxation processes. Data fitting using Equation (3) gives $B_1$ = 5.87(2) $\times$ $10^2$ $s^{-1}$, $B_2$ = 4.08(4) $\times$ $10^{-6}$ $Oe^{-2}$, $A_{Direct}$ = 2.06(3) $\times$ $10^3$ $T^{-2} \cdot s^{-1} \cdot K^{-1}$ and $B_4$ = 5.38(5) $\times$ $10^2$ $s^{-1}$. Note that the field dependence of the relaxation time cannot be described without a direct mechanism. The value of the parameter $A_{Direct}$ obtained from the $\ln(\tau)$ vs. inverse temperature dependence fit is close to the value obtained from $\tau$ vs. field dependence fit. The value of the inverse time $\tau^{-1} = B_4$ of field-independent magnetic relaxation processes obtained from $\tau$ vs. field dependence fit differs from the value $\tau^{-1}$ (Orbach and Raman) = 7.05(6) $\times$ $10^2$ $s^{-1}$ and is close to the value $\tau^{-1}$ (Orbach) = 5.66(6) $\times$ $10^2$ $s^{-1}$ obtained from the $\ln(\tau)$ vs. inverse temperature dependence fits by Equations (1) and (2), respectively. Thus, most likely, magnetic relaxation occurs through a combination of the Orbach and direct mechanisms.

The magnetization barrier value is somewhat higher than for K[Er(hfac)$_4$] [25]. The observed differences in the frequency dependences of $\chi''$ and magnetization barrier values of these two complexes are probably due to the existence of a strong dipole–dipole interaction in the K[Er(hfac)$_4$] complex. The Er–Er distances in the structure of this compound are markedly shorter than in complex **5** (7.94 Å vs. 9.67). It is known that the dipole–dipole interaction could reduce the thermal energy barrier [24]. In this connection, it should be noted that the dysprosium anionic complex [Dy(hfac)$_4$]$^-$ with polymeric cation [Cu(hfac)(Nit-Ph-PyIm)]$^+$, in which the Dy–Cu distance is 7.52 Å, has a magnetization barrier almost two times lower than that of complex **5**.

### 3.3. Quantum Chemical Calculations

The CASSCF/RASSI + SO/SINGLE_ANISO calculation of complex **5** was carried out to understand the origin of the magnetic anisotropy of Er ions and their magnetic behavior. Taking into account the closeness of the geometric structure of the symmetry-inequivalent Er complexes in the crystal structure, consideration was carried out only for one of them. The eight lowest Kramer's doublets (KDs) and the *g*-tensor components calculated for **5** using the SINGLE_ANISO code are summarized in Table 2.

**Table 2.** The ab initio computed energy levels (cm$^{-1}$) and associated *g*-tensors of the eight lowest KDs for Er ion in **5**.

| KD | Energy | $g_x$ | $g_y$ | $g_z$ |
|----|--------|-------|-------|-------|
| 1 | 0.0 | 0.665 | 0.831 | 14.376 |
| 2 | 45.3 | 1.094 | 5.346 | 12.578 |
| 3 | 82.4 | 2.965 | 3.512 | 6.319 |
| 4 | 110.8 | 0.058 | 0.087 | 16.935 |
| 5 | 224.9 | 2.801 | 3.240 | 9.374 |
| 6 | 277.0 | 3.407 | 4.414 | 6.198 |
| 7 | 315.8 | 0.143 | 0.431 | 10.490 |
| 8 | 349.0 | 0.296 | 0.375 | 9.209 |

The calculated effective $g_z$ component of the *g*-tensor is 14.376 for the ground KD of the Er(III) ion in **5** at $g_x$ and $g_y$ not equal to zero, which does not fully correspond to the Ising type feature and a pure $M_J$ = 15/2 ground state. This correlates with wave function decomposition analysis, which shows that the ground KD, in addition to the main contribution of 87.7%|±15/2>, contains a significant admixing from the 10.7%|±5/2 > state.

The analysis of magnetic relaxation pathways on the basis of transition magnetic moments (Figure 6) shows that in the case of the small energy gap between ground and first excited states, the strong mixing of the wave functions of the ground and first excited KDs, and also within the high values of the QTM probabilities, the relaxation should be through the QTM mechanism. In this regard, the complex under consideration does not exhibit slow magnetic relaxation in a zero magnetic field, and only in an applied field, when QTM is almost completely suppressed, can this process be observed.

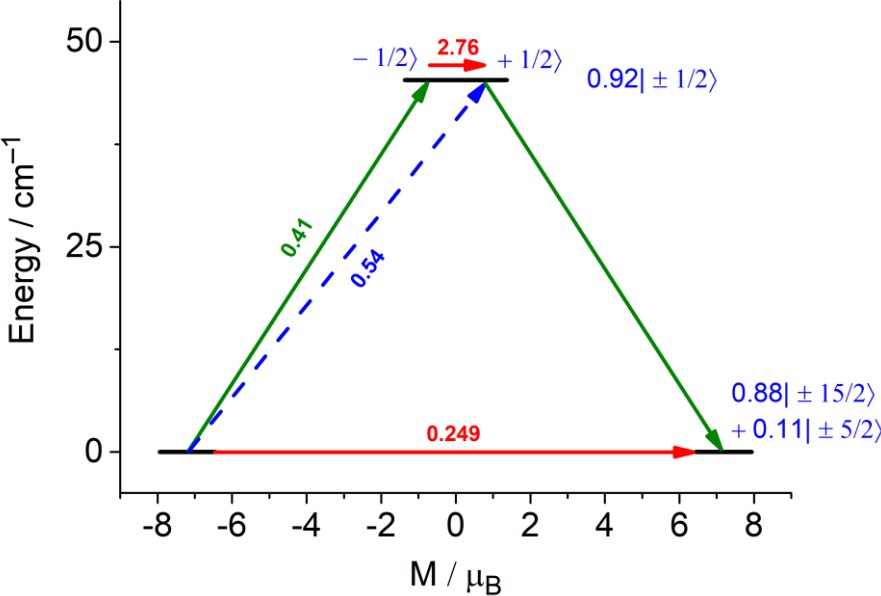

**Figure 6.** Computed possible magnetization relaxation pathways for Er(III) ions in **5**. The red arrows show QTM and TA-QTM via ground and higher-excited KD, respectively. The blue arrows show the Orbach process for relaxation. The green arrows show the mechanism of magnetic relaxation.

## 4. Conclusions

An anionic Er complex with the β-diketone ligands and an organic ammonium cation as the counterion was synthesized: $(CH_3)_4N[Er(hfac)_4]$, where hfac = hexafluoroacety-lacetonate). Its crystal structure and DC and AC magnetic properties were investigated.

The complex is the third in the row of Er complexes containing the $[Er(\beta\text{-diketone})_4]^-$ structural motif whose magnetic properties were studied and the first one with an organic cation. The Er(III) ions are in an eight-coordinated environment formed by eight oxygen atoms from four hfac ligands. According to the SHAPE analysis, the geometry around each independent Er(III) center can be described as a slightly distorted square antiprism (pseudo-$D_{4d}$ symmetry). The Er$\cdots$Er distances in the structure of the complex are 9.67 Å, which are noticeably greater than the closest Er$\cdots$Er distances (7.94 Å) in the $K[Er(hfac)_4]$ complex, which has a similar pseudo-$D_{4d}$ local symmetry. An AC-magnetic susceptibility study showed that the complex demonstrates a slow magnetic relaxation at an applied magnetic DC field, i.e., it is a field-induced single-ion magnet (SIM). In contrast to the $[Er(hfac)_4]^-$ complex with $K^+$ cation, which shows the frequency dependence of $\chi''$ in a constant field of 1000 Oe in a narrow range of temperatures (1.8–3.0 K) and frequencies near 1000 Hz only, the synthesized complex exhibits distinct $\chi''$ maxima over a wide range of temperatures (3–10 K) and frequencies (10–10,000 Hz). Quantum chemical calculations were performed to analyze the magnetic properties of the complex under consideration, which revealed agreement with the experimental results. The present study proposes a way for fine-tuning the magnetic dynamics of anionic $[Er(\beta\text{-diketone})_4]^-$ SIMs with pseudo-$D_{4d}$ symmetry by varying the organic ammonium cations remote from the Er(III) spin centers.

**Supplementary Materials:** The following supporting information can be downloaded at: https://www.mdpi.com/article/10.3390/magnetochemistry9060159/s1, Figure S1: Experimental (blue) and simulated from single-crystal data (red) powder X-ray diffraction pattern of the polycrystalline sample **5**; Figure S2: Projections of the fragments of the crystal structure of **5** on the *bc* and *ab* crystallographic planes; Figure S3: Plot of the $1/\chi_M$ vs. T. The red solid line represents the extrapolation of the high-temperature data to $1/\chi_M = 0$. The arrow marks the Weiss temperature; Figure S4: M vs. $BT^{-1}$ plots at 2, 3 and 5 K; Figure S5: Temperature dependences of the saturation magnetization and coercive field. The solid lines show the splines; Figure S6: Hysteresis loops at different magnetic field sweep rates (a) and temperatures (b). The inset shows low-field fragments of loops; (c) Dependence of the coercive field on the magnetic field sweep rate at a temperature of 2 K. Figure S7: Frequency dependences of the in-phase (a) and out-of-phase (b) AC susceptibility; (c) Cole–Cole plots for complex **5** at 4 K and indicated DC fields; (d) field dependence of the relaxation time τ at 4 K. Symbols are experimental data, solid lines indicate fits.; Table S1: Selected bond lengths [Å] and angles [deg] for **5**; Table S2: Shape analysis for different Er(III) ions in **5**; Table S3: The shortest intermolecular contacts; Table S4: Fit parameters for **5** at 4 K and indicated DC fields ($R^2 = 0.99913$); Table S5: Fit parameters for **5** at 1000 Oe DC fields and indicated temperatures ($R^2 = 0.99968$).

**Author Contributions:** Conceptualization and description sections: Introduction, Conclusion and Review, E.B.Y.; Synthesis of crystals and writing section: Synthesis, T.G.P.; The X-ray experiments, analysis and description of the X-ray data, D.V.K., G.V.S., A.I.D. and M.V.Z.; Analysis and writing section: Magnetic properties, D.V.K. and A.I.D.; Quantum chemical calculations, D.V.K. All authors have read and agreed to the published version of the manuscript.

**Funding:** This work has been supported by the state assignment of the Ministry of Science and Higher Education of the Russian Federation (No. AAAA-A19-119092390079-8). This research received no external funding.

**Institutional Review Board Statement:** Not applicable.

**Informed Consent Statement:** Not applicable.

**Data Availability Statement:** The data presented in this study are available on request from the corresponding author.

**Acknowledgments:** EPMA of the samples was carried out using the equipment of the Federal Research Center of Problems of Chemical Physics and Medicinal Chemistry RAS, http://www.icp.ac.ru/en (accessed on 10 October 2022).

**Conflicts of Interest:** The authors declare no conflict of interest. The funders had no role in the design of this study; in the collection, analyses, or interpretation of data; in the writing of the manuscript; or in the decision to publish the results.

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
