# Peer review of "The Organic Ammonium Counterion Effect on Slow Magnetic Relaxation of the [Er(hfac)4] Complexes"

_magnetochemistry, doi:10.3390/magnetochemistry9060159_

Round 1

Reviewer 1 Report

In this paper, the authors report a monoclear Er(III) complex with hfac ligands. The magnetic studies indicate that this Er complex exhibits field induced magnetic relaxation behavior. This work is interesting and will attract the readers working in this field. However, a few points, as pointed out below, are to be addressed prior publication.

1.For PXRD, the experimental PRXD does not well match with the theoretical one.

2. For prolate Er(III) ion, the ideal coordination environment to possess high anisotropy is equatorial coordination. Thus, the coordination geometry of Er(III) should be discussed.

3. In Figures 1 and 3, “ER”? page 9, line 268, “31.2 K”? It should be 45.3 cm-1(see Table 2), i.e. 65.2 K.

Author Response

Dear Editor,

We are grateful to the reviewers for their valuable comments, which have helped us to improve the article. The manuscript is corrected accordingly. Below we provide our responses to the comments made by the reviewers.

    Reviewer 1

In this paper, the authors report a monoclear Er(III) complex with hfac ligands. The magnetic studies indicate that this Er complex exhibits field induced magnetic relaxation behavior. This work is interesting and will attract the readers working in this field. However, a few points, as pointed out below, are to be addressed prior publication.

1.For PXRD, the experimental PRXD does not well match with the theoretical one.

We have performed the PXRD experiment on the same sample, in our opinion, it turned out to be a good match with the simulated one from single crystal data, Fig. S1 is updated.

  1. For prolate Er(III) ion, the ideal coordination environment to possess high anisotropy is equatorial coordination. Thus, the coordination geometry of Er(III) should be discussed.

We have added a more detailed description of the coordination environment of Er(III) ions to the text of the manuscript.

  1. In Figures 1 and 3, “ER”? page 9, line 268, “31.2 K”? It should be 45.3 cm-1(see Table 2), i.e. 65.2 K.

We have corrected Fig.1 and 2.

Indeed, this unfortunate typo was made when converting «cm-1» to «K» incorrectly. Instead of multiplying, the division by 1.44 was performed. This discrepancy has been removed from the text of the manuscript.

Reviewer 2 Report

Prokhorova et al. report a novel mononuclear anionic erbium complex [Er(hfac)4]- (hfac = hexafluoroacetylacetone) with an organic ammonium cation [(CH3)4N+] as the counterion. The molecular structure and magnetic properties of this compound are well-performed. More importantly, the experimental study of the magnetic properties has been complemented by theoretical analysis based on ab initio calculations. This work is simple and clear. The material design strategy and chemical characterizations are convinced. Therefore, I recommend a minor revision. Some detailed comments are listed as follows:

1. On the crystal structure, the authors should draw a packing mode of the compound. And, the hydrogen bonds of C-H…O and C-H…F should be discussed in the MS and listed in the SI.

2. The Er-O and O…O distance is not enough to describe the difference in Er1, Er2 and Er3, the angle between the diagonals of the two squares and the angle between the S8 axis and a Dy–O vector should be discussed.

3. The reason for the difference between the K/Cs-compound and the [(CH3)4N+]-compound should be discussed.

4. There are some unrecognizable symbols in Figure 4 and Figure 5.

5. There are two Figure 3 in the MS, please check. Some minor style errors should be carefully revised.

Author Response

Dear Editor,

We are grateful to the reviewers for their valuable comments, which have helped us to improve the article. The manuscript is corrected accordingly. Below we provide our responses to the comments made by the reviewers.

Reviewer 2

Prokhorova et al. report a novel mononuclear anionic erbium complex [Er(hfac)4]- (hfac = hexafluoroacetylacetone) with an organic ammonium cation [(CH3)4N+] as the counterion. The molecular structure and magnetic properties of this compound are well-performed. More importantly, the experimental study of the magnetic properties has been complemented by theoretical analysis based on ab initio calculations. This work is simple and clear. The material design strategy and chemical characterizations are convinced. Therefore, I recommend a minor revision. Some detailed comments are listed as follows:

  1. On the crystal structure, the authors should draw a packing mode of the compound. And, the hydrogen bonds of C-H…O and C-H…F should be discussed in the MS and listed in the SI.

We have added figures and discussion of the crystal packing mode for compound under consideration to the text of the manuscript.

  1. The Er-O and O…O distance is not enough to describe the difference in Er1, Er2 and Er3, the angle between the diagonals of the two squares and the angle between the S8 axis and a Dy–O vector should be discussed.

We have added a more detailed description and comparison of the coordination environment of Er(III) ions using additional parameters of SAP polyhedron to the text of the manuscript.

  1. The reason for the difference between the K/Cs-compound and the [(CH3)4N+]-compound should be discussed.

In the text of the manuscript there is a discussion of the reason* for the difference in the properties of the [Er(hfac)4]- anion with different cations. The difference between K+ and Cs+ salts was established earlier in [25] and explained by the difference in the coordination environment of Ln(III) ions; this is pointed in the Introduction.

* «The magnetization barrier value is somewhat higher than for the K[Er(hfac)4] [25]. The observed differences in the frequency dependences of χ″ and magnetization barrier values of these two complexes are probably due to the existence of a strong dipole-dipole interaction in the K[Er(hfac)4] complex. The Er-Er distances in the structure of this compound are markedly shorter than in complex 5 (7.94 Å vs 9.67). It is known that the dipole-dipole interaction could reduce the thermal energy barrier [43]. In this connection, it should be noted that the dysprosium anionic complex [Dy(hfac)4]- with polymeric cation [Cu(hfac)(Nit-Ph-PyIm)]+, in which the Dy-Cu distance is 7.52 Å, has a magnetization barrier almost two times lower than that of complex 5

  1. There are some unrecognizable symbols in Figure 4 and Figure 5.

We have corrected Fig.4 and 5.

  1. There are two Figure 3 in the MS, please check. Some minor style errors should be carefully revised.

We have corrected this typo.

Reviewer 3 Report

The manuscript describes the synthesis, structural and magnetic characterization of an Er(III) complex with the well-known ligand hfac-: [Er(hfac)4]-. Although this complex was already known as their K+ and Cs+ salts, authors have prepared the salt with the tetramethylammonium cation to study the influence of the cation on the magnetic properties. The title compound shows a field-induced single-ion magnet behavior that has been well characterized. The work is well performed, including theoretical ab-initio calculations, and the results are interesting for the community working in SIM and SMM and, therefore, I consider that the manuscript can be accepted for publication after some minor revision as follows:

1.     It is surprising to see such a strong antiferromagnetic coupling between the Er(III) ions located almost 10 Å apart since 4f electrons are very deep and dipolar interactions are expected to be almost negligible. Most probably, the decrease observed in the XT product when the temperature is decreased, must be attributed to the depopulation of the excited levels (mJ) arising due to the ligand field splitting. This decrease leads to negative non negligible theta value even if there are no Er-Er interactions at all. In fact, this interpretation is later used to justify the lower saturation value of the isothermal magnetization.

2.     The presence of a hysteresis loop is also surprising because for zero DC field there is a fast relaxation of the magnetization, as shown by the AC measurements. When one sees the dependence of the coercive field with the scan rate of the magnetic field, it is clear that this is a phenomenon associated with the scan rate. Furthermore, if we plot the coercive field as a function of the scan rate, we can see the coercive field is almost equal to the scan rate. Therefore, for very low scan rates, the coercive field should be almost zero. I suggest to perform hysteresis measurements with lower scan rates and extrapolate the coercive field for a scan rate = 0 Oe/s.

3.     Please, indicate the parameters obtained  and the equation used to fit the field dependence of the relaxation time (figure S6.d).

4.     Given the almost linear dependence of the Arrhenius plot of the relaxation time, authors should also try to fit this plot with a model including Orbach + Direct mechanisms (instead of Orbach + Raman).

5.     Finally, a more detailed description of the theoretical calculations is missing, especially for the determination of the energy barrier value with the ab initio calculations.

Author Response

Dear Editor,

We are grateful to the reviewers for their valuable comments, which have helped us to improve the article. The manuscript is corrected accordingly. Below we provide our responses to the comments made by the reviewers.

Reviewer 3

The manuscript describes the synthesis, structural and magnetic characterization of an Er(III) complex with the well-known ligand hfac-: [Er(hfac)4]-. Although this complex was already known as their K+ and Cs+ salts, authors have prepared the salt with the tetramethylammonium cation to study the influence of the cation on the magnetic properties. The title compound shows a field-induced single-ion magnet behavior that has been well characterized. The work is well performed, including theoretical ab-initio calculations, and the results are interesting for the community working in SIM and SMM and, therefore, I consider that the manuscript can be accepted for publication after some minor revision as follows:

  1. It is surprising to see such a strong antiferromagnetic coupling between the Er(III) ions located almost 10 Å apart since 4f electrons are very deep and dipolar interactions are expected to be almost negligible. Most probably, the decrease observed in the XT product when the temperature is decreased, must be attributed to the depopulation of the excited levels (mJ) arising due to the ligand field splitting. This decrease leads to negative non negligible theta value even if there are no Er-Er interactions at all. In fact, this interpretation is later used to justify the lower saturation value of the isothermal magnetization.

We agree with the comment and have amended the article accordingly: the value of Weiss constant must be treated with caution, since it can be caused not only by exchange or dipole interaction, but the depopulation of the thermal excited levels arising due to the crystal-field splitting. This decrease leads to negative non negligible Weiss constant even if there are no magnetic moment interactions.

  1. The presence of a hysteresis loop is also surprising because for zero DC field there is a fast relaxation of the magnetization, as shown by the AC measurements. When one sees the dependence of the coercive field with the scan rate of the magnetic field, it is clear that this is a phenomenon associated with the scan rate. Furthermore, if we plot the coercive field as a function of the scan rate, we can see the coercive field is almost equal to the scan rate. Therefore, for very low scan rates, the coercive field should be almost zero. I suggest to perform hysteresis measurements with lower scan rates and extrapolate the coercive field for a scan rate = 0 Oe/s.

We agree with the comment. Indeed, the observed hysteresis is most likely related to the magnetic field sweep rate. Corresponding additions have been introduced in the text of the paper: the presence of a magnetic hysteresis loop is surprising because for zero DC field there is a fast relaxation of the magnetization, as shown below by the AC measurements. Magnetic hysteresis measurements with different magnetic field sweep rate were performed. We can see that coercive field is almost equal to the scan rate (Figure S5c, ESI). If we plot dependence of the coercive field on the magnetic field sweep rate and extrapolate the coercive field to very low sweep rates (0 Oe/s), the coercive field becomes almost zero (Figure S5c, ESI). Thus, the magnetic hysteresis is most likely related to the magnetic field sweep rate.

  1. Please, indicate the parameters obtained and the equation used to fit the field dependence of the relaxation time (figure S6.d).

Approximation of the dependence of ln(τ) vs inverse temperature does not allow one to give an unambiguous answer according to which of the mechanism relaxation (Raman or Direct) occurs. Therefore, fit of the field dependence of the relaxation time was performed with Eq(3) (Fig.S6d, ESI).

τ–1 =

B1/(1+B2H2)

+ AH2T

+ B4

(3)

QTM

Direct

 Orbach and Raman

Where the first two terms correspond to the field-dependent quantum tunneling and direct mechanisms of magnetic relaxation. Third term corresponds to the filed-independent Orbach and Raman magnetic relaxation processes. Data fitting using the Eq(3) gives for B1 = 5.87(2)·102 s−1, B2 = 4.08(4)·10−6 Oe−2, ADirect = 2.06(3)·103 T−2·s−1·K−1 and B4 = 5.38(5)·102 s−1. Note that the field dependence of the relaxation time cannot be described without a direct mechanism. The value of the parameter ADirect obtained from the ln(τ) vs inverse temperature dependence fit is close to the value obtained from τ vs field dependence fit. The value of the inverse time τ−1 = B4 of field-independent magnetic relaxation processes obtained from τ vs field dependence fit differs from the value τ−1 (Orbach and Raman) = 7.05(6)·102 s−1 and is close to the value τ−1 (Orbach) = 5.66(6)·102 s−1 obtained from the ln(τ) vs inverse temperature dependence fits by equations (1) and (2), respectively. Thus, most likely, magnetic relaxation occurs through a combination of the Orbach and direct mechanisms.

  1. Given the almost linear dependence of the Arrhenius plot of the relaxation time, authors should also try to fit this plot with a model including Orbach + Direct mechanisms (instead of Orbach + Raman).

Fit of the experimental dependence of the ln(τ) vs inverse temperature was performed also with Eq(2), including Orbach and direct mechanisms. Data fitting using the Eq(2) gives a higher energy barrier of 28.54(8) K, with a lower pre-exponential factor τ0 of 1.40(3)·10−6 s and ADirect = 2.38(8)·103 T−2·s−1·K−1. As can be seen from the Fig. 5b, the quality of approximations by both equations (1) and (2) is the same and gives close values of the energy barrier and pre-exponential factor for Orbach component.

  1. Finally, a more detailed description of the theoretical calculations is missing, especially for the determination of the energy barrier value with the ab initio calculations.

Such ab initio calculations are no longer so unique. A detailed description of the procedure is given in the experimental part with references to fundamental works. As for the determination of the energy barrier value, in this case, a comparison is made with the energies of the excited states relative to the ground state, most often, for systems with low symmetry (C1), they are limited only to the first excited state.